# PrEP knowledge and perceptions among women living in North Carolina public housing communities

**Lauren M. Hill**[1,2*], Olivia Allison[1,2], Oluwamuyiwa Adeniran[1,2], Marcella Jones[2],
**Suur Ayangeakaa**[2,3,4], Tonya Stancil[1,2], K. Jean Phillips-Weiner[1,2], Alexandra F. Lightfoot[1,2],
**Mehri S. McKellar**[2,5], Carol E. Golin[1,2,6]

1 Department of Health Behavior, Gillings School of Global Public Health, University of North Carolina at Chapel Hill, Chapel Hill, North Carolina, United States of America, 2 IFE Community Academic Partnership, Durham, North Carolina, United States of America, 3 Department of Population Health Sciences, Duke University School of Medicine, Durham, North Carolina, United States of America, 4 Duke Global Health Institute, Duke University, Durham, North Carolina, United States of America, 5 Department of Medicine, Duke University School of Medicine, Durham, North Carolina, United States of America, 6 Department of Medicine, University of North Carolina at Chapel Hill, Chapel Hill, North Carolina, United States of America

* hilllm@email.unc.edu

## Abstract

Women in low-income communities are disproportionately affected by HIV yet have been largely left out of efforts to raise awareness about pre-exposure prophylaxis (PrEP). To inform future awareness campaigns, we assessed women's current knowledge of and attitudes toward PrEP. We surveyed 184 women living in public housing communities in North Carolina regarding PrEP knowledge, attitudes, and perceived norms, as well as reported HIV-associated factors and perceived HIV acquisition chances. 38 women participated in eight focus group discussions (FGDs) addressing personal and community PrEP perceptions. Survey participants were 46 years old on average, and 86% identified as Black/African American. Only 35% had heard of PrEP, yet, after being told what it was, 61% said they probably or definitely would take PrEP in the next 6 months. Most women believed that if they decided to take PrEP, their partner (72%) or their family (66%) would approve. When asked about the importance of factors influencing their interest in PrEP, women most frequently rated possible side effects as important or very important (76%), followed by cost considerations (67% for cost of PrEP, 74% for cost of clinic visits and labs). In the FGDs, women had limited PrEP knowledge, but several had heard of PrEP from television commercials, which gave them the impression that PrEP was only for men. Women were concerned about potential side effects, interactions with other medications, safety during pregnancy, and the burden of daily dosing. Most FGD participants expressed generally positive attitudes toward PrEP, but some thought other women would be uninterested due to low perceived chances of HIV acquisition. Overall, these results suggest that while few women had previously heard of PrEP, most were interested in PrEP after receiving information about it and perceived positive community attitudes toward PrEP. Our findings indicate the importance of community-based PrEP

**Data availability statement:** All data files are available from the ICPSR database (https://www.openicpsr.org/openicpsr/project/218062/version/V1/view).

**Funding:** This research was funded by Gilead Sciences. Additional investigator and administrative support were provided by the National Institute of Mental Health (K01 MH121186) and the National Institute of Allergy and Infectious Disease (P30 AI050410). The funders had no role in study design, data collection and analysis, decision to publish, or preparation of the manuscript.

**Competing interests:** I have read the journal's policy and the authors of this manuscript have the following competing interests: This study was funded by Gilead Sciences. CEG and SA report receiving consultant fees from Gilead Sciences for activities not directly related to this work.

communication that speaks to cisgender women, provides information on side effects, and offers destigmatized messaging regarding reasons for HIV prevention for women to consider.

## Introduction

African American cisgender women in the South are disproportionately affected by HIV infection yet have inequitable access to pre-exposure prophylaxis (PrEP) for HIV prevention [1–7]. Cisgender women make up a quarter of people living with HIV and one in five new HIV diagnoses in the US [8,9]. In particular, Black or African American women in the Southeast are disproportionately affected by HIV [3–7,10,11]. Black women make up 13% of the female population but account for 54% of new HIV cases in women (and 57% of prevalent cases) [12], a much higher rate than any other racial or geographic group among women [12]. Women living in the Southeast subregion also face the highest burden of HIV in the nation [12]. PrEP holds immense potential to reduce the overall burden of HIV in the US and address HIV-related disparities, yet it remains underutilized, especially among women. The CDC estimates that women constitute over 15% of individuals with PrEP indications in the US [13], yet make up less than 5% of PrEP users [14]. Only an estimated 2% of the 170,000 women who could benefit from PrEP are using it [15]. Gender disparities in PrEP use are particularly severe for Black women, who make up more than half of new infections among women but only a quarter of the low number of female PrEP users [14]. Disseminating PrEP as a prevention method for cisgender women is particularly urgent because it can be controlled by the receptive partner, overcoming the gender-based difficulties that women can face with condoms [16–18].

In the South, disparities in PrEP uptake may partly stem from social and economic structures such as limited healthcare access, low insurance coverage, poverty, economic inequality, historical segregation [19–22], and pervasive community HIV stigma [23–27]. Structural racism is a major driver of these factors in the United States, with discriminatory beliefs and systems driving disparities in many of the environmental factors critical for health including housing, health care, education, employment, and criminal justice [28–30]. In addition, side effects and cost [31–33], as well as HIV-related stigma may also pose barriers to PrEP uptake among women [34,35].

African American women living in low-income neighborhoods have higher chances of acquiring HIV than the general population of African American women living in the US [36], likely because they experience the intersection of multiple structural barriers to accessing PrEP. These barriers include limited transportation, lack of childcare, insufficient financial resources, challenges in navigating the healthcare system, and medical mistrust [1,2,30]. PrEP awareness is also an issue; fewer than 15% of women with reasons for HIV prevention are aware of PrEP [37,38]. PrEP awareness and use may be low among cisgender women due to marketing typically targeting men who have sex with men (MSM) and transgender women [39,40]. Because of this marketing, many women who are aware of PrEP believe that it is not recommended for use by cisgender women [41]. Yet, PrEP acceptability has been very high among women once they are made aware of it [42,43].

To promote awareness of PrEP by women living in low-income neighborhoods in North Carolina (NC), we conducted a study to develop a PrEP communication strategy for women living in public housing. Although online campaigns focused on cisgender women in the US, such as the CDC's #ShesWell initiative [44], have been implemented, there are few

evidence-based approaches specifically designed for community-level communication target-ing cisgender women in development [45]. In the formative study reported here, we assessed PrEP knowledge, perceptions, and norms, as well as HIV-associated factors and perceptions among women living in public housing communities in a mid-sized city in North Carolina.

## Materials and methods

### Study context

The data collection presented in this paper was undertaken in the formative phase of the IFE4Her study to inform development of a community-based intervention to raise PrEP awareness among women living in public housing communities and deliver PrEP services in the community in a mobile medical unit. This study was conducted with a longstanding formal collaborative Community Academic Partnership (CAP) of public housing commu-nity residents, public housing staff members, and other interested community members and community-based organization representatives. Study investigators have collaborated with the CAP for over a decade to promote sexual health among public housing residents in a mid-sized city in North Carolina. As demonstrated by HPTN 064, these communities coincide with low-income census tracts that have notably high rates of HIV cases [46], and the city where we conducted the study ranked fourth among NC counties in the number of newly diagnosed HIV cases in 2014-2016 [47]. To develop this intervention, we conducted a commu-nity survey and focus group discussions with women living in public housing communities. This formative study was completed in 2021 and 2022; as only TDF/FTC had been approved for use by cisgender women at that time, all our questions regarding PrEP concerned only TDF/FTC daily oral PrEP.

### Participant recruitment

Women were eligible to participate in the formative study if they met the following crite-ria: 18 years of age or older; identify as cisgender female (assigned female sex at birth and currently identify as female); and resided in public housing at the time of the study. Eligible women could participate in the survey, a focus group discussion (FGD), or both. Women were recruited from the community on a convenience basis and enrolled separately for each activity they participated in (survey and/or FGD). Convenience sampling was selected because a resident list or contact information from which to draw a probabilistic sample was not available. We recruited participants by tabling in public housing communities, speaking with residents face-to-face, posting flyers throughout the community, and through distribution of flyers to residents by public housing staff and study participants who shared flyers with other residents. We recruited for the survey in all public housing communities. For the focus groups, we purposively recruited residents in the four public housing communities that we planned to include in a future pilot study of the IFE4Her intervention.

### Quantitative data collection

We conducted the community survey with adult women living in public housing commu-nities from February 21, 2022 to August 31, 2022. Participants self-administered an elec-tronic Qualtrics survey either using a study tablet or on their own device using a QR code from the recruitment flyers. To assess PrEP knowledge, women responded to single items regarding previous knowledge of PrEP and sources of information, personal experience with and knowledge of others using PrEP, and the likelihood of using PrEP in the next six months if affordable. To characterize the potential PrEP need among participants, we

assessed HIV-associated factors per CDC PrEP criteria [48], including partner HIV status, consistent condom use, STI history, and injection drug use. To assess partner HIV status and consistency of condom use, women were asked questions regarding each of their three most recent male sexual partners in the past year. For each partner, they reported their partner's HIV status if known, and the frequency of condom use with their partner in the most recent month they had sex (1-all of the time to 5-none of the time). For reporting, responses other than using condoms "all of the time" were coded as a report of inconsistent condom use. Women also responded to single items regarding their HIV testing history and status, number of sexual partners in the past year, their perceived likelihood of acquiring HIV in the next year (1-very unlikely to 5-very likely), and their level of concern about acquiring HIV (1-no concern at all to 5-very concerned). Participants responded to questions regarding perceived community norms regarding PrEP (anticipated approval by others) by rating statements on a five-point Likert scale (1-strongly disagree to 5-strongly agree). Items included covered all primary referent groups for social norms including romantic partners, family, friends, and community members. Participants also responded to questions regarding factors influencing PrEP interest by rating statements on a five-point Likert-type scale (1-very unimportant to 5-very important). Factors included were identified in the literature as driving PrEP interest among women and included considerations regarding cost, side effects, privacy, and daily pill burden.

## Qualitative data collection

We conducted 8 focus groups in four public housing communities between December 10, 2021 and July 23, 2022. Focus group discussions were conducted with between 3 to 8 participants in private meeting rooms in the community. Participants individually completed a brief sociodemographic questionnaire prior to initiating the FGD. FGDs were facilitated by a trained research staff member and a note-taker and lasted approximately two hours. Key FGD topics included: PrEP knowledge, likes and dislikes about PrEP, and perceived appropriateness of PrEP for women in the community. Following questions about PrEP knowledge, participants were shown a brief CDC video that provided basic knowledge about daily oral PrEP including on how it works, its efficacy, side effects, and how to get it. Focus group discussions were digitally audio recorded and transcribed verbatim for analysis with identifying information redacted.

## Ethical considerations

All study procedures were approved by the institutional review boards of the University of North Carolina and Duke University. All survey participants completed electronic consent prior to questionnaire completion. All focus group participants completed written consent before participation.

## Data analysis

Survey results were summarized with descriptive statistics using SAS v9.4. All statistics generated were univariate except for bivariate comparisons to examine perceived HIV acquisition chances and PrEP interest relative to HIV-associated factors. To draw these comparisons, we first transformed the measures of interest to dichotomous variables. We created a dichotomous HIV-associated factors variable categorizing participants who reported any HIV-associated factor per CDC criteria [48], and those who reported none (1 = one or more factors reported; 0 = no factors reported). To dichotomize perceived HIV acquisition chances, participants were scored as 1 if they reported that the likelihood of HIV infection in the next

year is "likely" or "very likely," and women with any other response as 0. To dichotomize HIV-acquisition concern, participants were scored as 1 if they reported that they are "somewhat concerned" or "very concerned" about their chances of HIV infection, and participants with any other response as 0. Finally, to dichotomize PrEP interest, participants were scored as 1 if they reported that they "probably would" or "definitely would" use PrEP in the next 6 months if affordable, and any other response as 0. We then cross-tabulated reporting of HIV-associated factors with perceived HIV acquisition chances, HIV-acquisition concern, and PrEP interest, respectively (reporting proportions and percentages of each variable by number of HIV-associated factors reported) and calculated a Chi-square test statistic and p-value for each comparison.

Qualitative analysis using deidentified transcripts consisted of: 1) Reading transcripts until content became intimately familiar. Emergent themes were noted as transcripts were reviewed; 2) Coding using a structured codebook based on identified themes in addition to codes corresponding to interview guide topics. Two trained coders applied codes to transcripts using Dedoose qualitative software. To ensure inter-coder reliability, 10% of data were double-coded; 3) Data reduction through review of the data related to each code to identify principal sub-themes reflecting finer distinctions in the data, observing the variation and richness of themes identified, and noting differences between participants and focus groups; 4) Data display and comparison using matrices to categorize and display data to help facilitate comparisons across focus groups.

## Results

### Survey results

Survey participants were 46 years old on average (range: 18-84), and the majority (83%) identified as African American or Black non-Hispanic (Table 1). The majority had a high school diploma or less (64%) and an annual household income of less than $10,000 (62%). Most participants had public health insurance (41% Medicaid, 26% Medicare).

We assessed HIV-associated factors and participants' perceived HIV acquisition chances (Table 2). Most participants reported having tested for HIV (84%); none of the women who had tested reported an HIV-positive result. When asked about HIV-associated factors, 46% of participants reported ever having an STI diagnosis, and 9% reported inconsistent condom use with one or more partners of unknown HIV status in the past year. No participant reported a partner living with HIV whose viral load was unknown or detectable (also, no participant reported any partner living with HIV), nor did any participant report injection drug use with equipment sharing. Altogether, 55% of respondents reported any HIV-associated factors. Regarding perceptions (Table 3), a large majority believed that their likelihood of acquiring HIV in the next year was very unlikely or unlikely (90%). About half (56%) reported not being concerned at all about acquiring HIV.

When asked about their prior knowledge of PrEP (Table 4), 35% reported having heard of PrEP. Among those who had heard of it, the most common sources of PrEP information were from television commercials and healthcare providers. Eleven women (7%) reported knowing anyone who had used PrEP and three women (2%) reported using PrEP. Ninety-eight women (61%) said they probably or definitely would take PrEP in the next 6 months if it were affordable after learning about what it was.

We also examined perceived HIV acquisition chances, HIV acquisition concerns, and PrEP interest relative to reporting of HIV-associated factors (Table 5). Comparing women with any reported HIV-associated factor to those with none, 10% and 0% respectively reported moderate or high perceived HIV acquisition chances; 32% and 18% respectively reported moderate

**Table 1. Survey participant demographic characteristics.**

| Variable | N | n (%) or mean [SD] |
|---|---|---|
| **Age (years)** | 184 | 46.05 [15.79] |
| **Race/ethnicity** | 176 | |
| African American or Black non-Hispanic | | 146 (82.95%) |
| African American or Black Hispanic | | 5 (2.84%) |
| White non-Hispanic | | 6 (3.41%) |
| Multiple races | | 8 (4.55%) |
| Other | | 11 (6.25%) |
| **Relationship status** | 169 | |
| Single, never married | | 75 (44.38%) |
| Married | | 16 (9.47%) |
| In a relationship with a steady partner | | 37 (21.89%) |
| Widowed, separated, or divorced | | 36 (21.30%) |
| Other | | 5 (2.96%) |
| **Highest level of education** | 170 | |
| Less than high school | | 40 (23.53%) |
| High school diploma or equivalent | | 69 (40.59%) |
| Post-secondary education, no bachelor's | | 51 (30.00%) |
| Bachelor's Degree or higher | | 10 (5.88%) |
| **Household income** | 152 | |
| 0 to $4,999 | | 76 (50.00%) |
| $5,000 to $9,999 | | 18 (11.84%) |
| $10,000 to $14,999 | | 25 (16.45%) |
| $15,000 or more | | 33 (21.71%) |
| **Insurance status** | 164 | |
| Uninsured | | 34 (20.73%) |
| Medicare | | 43 (26.22%) |
| Medicaid | | 67 (40.85%) |
| Private/commercial health insurance | | 8 (4.88%) |

**Table 2. Sexual relationships and HIV-associated factors.**

| Variable | N | n (%) or mean [SD] |
|---|---|---|
| **Ever tested for HIV** | 180 | 151 (83.89%) |
| **Male sexual partners in past year** | 159 | |
| 0 partners | | 34 (21.38%) |
| 1 partner | | 80 (71.70%) |
| 2 or more | | 45 (28.30%) |
| **Reported HIV-associated factors** | | |
| HIV positive partner with unknown or undetectable viral load | 142 | 0 (0%) |
| Inconsistent condom use with partner(s) of unknown HIV status | 133 | 12 (9.02%) |
| Ever had STI diagnosis | 157 | 72 (45.86%) |
| Injection drug use with equipment sharing | 173 | 0 (0%) |
| Any factor | 141 | 77 (54.61%) |

**Table 3. Personal HIV perceptions.**

| Variable | N | n (%) or mean [SD] |
|---|---|---|
| **Likelihood of acquiring HIV in next year** | 169 | |
| Very unlikely | | 118 (69.82%) |
| Unlikely | | 34 (20.12%) |
| About equally likely and unlikely | | 7 (4.14%) |
| Likely | | 3 (1.78%) |
| Very likely | | 7 (4.14%) |
| **Level of HIV-acquisition concern** | 167 | |
| Not at all concerned | | 93 (55.69%) |
| A little concerned | | 27 (16.17%) |
| Somewhat concerned | | 19 (11.38%) |
| Very concerned | | 28 (16.77%) |

**Table 4. PrEP knowledge, experience, and interest.**

| Variable | N | n (%) or mean [SD] |
|---|---|---|
| **Ever heard of PrEP** | 175 | 62 (35.43%) |
| **Where heard about PrEP** | 61 | |
| TV commercial | | 29 (47.54%) |
| Healthcare providers | | 26 (42.62%) |
| Public health department | | 14 (22.95%) |
| Family or friends | | 12 (19.67%) |
| Internet | | 10 (16.39%) |
| Social media | | 8 (13.11%) |
| HIV/AIDS organization | | 5 (8.20%) |
| Other | | 9 (14.75%) |
| **Know anyone who has used PrEP** | 169 | 11 (6.51%) |
| **Ever used PrEP** | 174 | 3 (1.72%) |
| **Likelihood of using PrEP in next 6 months if affordable** | 161 | |
| Definitely would not | | 35 (21.74%) |
| Probably would not | | 28 (17.39%) |
| Probably would | | 69 (42.86%) |
| Definitely would | | 29 (18.01%) |

**Table 5. Comparison of reported HIV-associated factors with perceived HIV acquisition chances, HIV acquisition concerns, and PrEP interest.**

| | HIV-associated factors reported | |
|---|---|---|
| | 0 | 1 or more |
| **Moderate or high perceived HIV acquisition chances**[a] (n = 138)[**] | 0 (0%) | 8 (10.39%) |
| **Moderate or high HIV acquisition concerns**[b] (n = 138) | 11 (17.74%) | 24 (31.58%) |
| **Likely to use PrEP in next 6 months if affordable**[c] (n = 129)[*] | 31 (50.82%) | 49 (72.06%) |

[a]*Reporting that the likelihood of HIV infection in the next year is "likely" or "very likely".*

[b]*Reporting that the that they are "somewhat concerned" or "very concerned" about their chances of HIV infection.*

[c]*Reporting that the that they "probably would" or "definitely would" use PrEP in the next 6 months if affordable.*

[*]*Indicates Chi-square p-value < .05 for differences in proportions of indicated variable by presence of HIV-associated factors.*

[**]*Indicates Chi-square p-value < .01 for differences in proportions of indicated variable by presence of HIV-associated factors*

or high concern about acquiring HIV; and 72% and 51% respectively reported that they would be likely to use PrEP in the next 6 months if affordable.

When asked about PrEP norms in the community (Table 6), women mostly agreed or strongly agreed with statements about community interest and approval or PrEP use. Regarding the extent to which others would approve if they decided to use PrEP, the greatest proportion of women endorsed the belief their partner(s) would approve (72% agree or strongly agree), followed by belief that their family would approve (66% agree or strongly agree), the belief that their friends would approve (62% agree or strongly agree), and the belief that people in the community would approve (56% agree or strongly agree). About half of participants (51%) endorsed the belief that women in their community would be interested in PrEP if they knew about it.

When asked about how important different factors would be in influencing their interest in using PrEP (Table 7), women most frequently rated possible side effects as important or very important (76%). Cost considerations were the next most important, with 67% and 74% saying the cost of PrEP and the cost of clinic visits, respectively, would be important or very important. Partners finding out and the need to take a pill every day were endorsed as important or very important by a little more than half of participants (59% and 57%, respectively). 47% of participants said family, friends, or community members finding out they take PrEP would be important or very important in their decision to use it.

## Focus group results

38 women participated in eight FGDs. FGD participants ranged in age from 19 to 67 years old, and all but one participant identified as African American or Black non-Hispanic (Table 8). Most participants (20) identified as single and never married and had a high school degree or less (25).

**Table 6.  Perceptions of community PrEP norms.**

| Item | N | Rating (n %) | | | | |
|---|---|---|---|---|---|---|
| | | Strongly Disagree | Disagree | Neither Agree nor Disagree | Agree | Strongly Agree |
| If I decided to use PrEP my partner would approve | 132 | 11 (8.33%) | 13 (9.85%) | 13 (9.85%) | 58 (43.94%) | 37 (28.03%) |
| If I decided to use PrEP, my family would approve | 168 | 12 (7.14%) | 10 (5.95%) | 35 (20.83%) | 76 (45.24%) | 35 (20.83%) |
| If I decided to use PrEP my friends would approve | 169 | 11 (6.51%) | 13 (7.69%) | 40 23 (23.67%) | 78 (46.15%) | 27 (15.98%) |
| If I decided to use PrEP, people in my community would approve | 169 | 9 (5.33%) | 13 (7.69%) | 53 (31.36%) | 70 (41.42%) | 24 (14.20%) |
| Women in my community would be interested in taking PrEP if they knew about it | 172 | 8 (4.65%) | 6 (3.49%) | 40 (23.26%) | 88 (51.16%) | 30 (17.44%) |

**Table 7.  Influences on interest in using PrEP.**

| Item | N | Rating (n %) | | | | |
|---|---|---|---|---|---|---|
| | | Very unimportant | Not important | Somewhat important | Important | Very Important |
| Cost of PrEP | 167 | 20 (11.98%) | 12 (7.19%) | 23 (13.77%) | 47 (28.14%) | 65 (38.92%) |
| Cost of clinic visits and tests | 167 | 18 (10.78%) | 9 (5.39%) | 16 (9.58%) | 55 (32.93%) | 69 (41.32%) |
| Possible side effects | 169 | 20 (11.83%) | 9 (5.33%) | 12 (7.10%) | 34 (20.12%) | 94 (55.62%) |
| Partner(s) finding out | 140 | 19 (13.57%) | 26 (18.57%) | 13 (9.29%) | 33 (23.57%) | 49 (35.00%) |
| Friends, family, or community finding out | 163 | 23 (14.11%) | 47 (28.83%) | 17 (10.43%) | 40 (24.54%) | 36 (22.09%) |
| The need to take a pill everyday | 166 | 12 (7.23%) | 15 (9.04%) | 48 (28.92%) | 78 (46.99%) | 15 (9.04%) |

**Table 8. Focus group participant demographics (N = 38).**

| Variable | n or median [range] |
|---|---|
| **Age (years)** | 50 [19-67] |
| **Race/ethnicity** | |
| African American or Black non-Hispanic | 37 |
| White non-Hispanic | 1 |
| **Relationship status** | |
| Single, never married | 20 |
| Married | 2 |
| In a relationship with a steady partner | 5 |
| Widowed, separated, or divorced | 6 |
| Other | 2 |
| **Highest level of education** | |
| Less than high school | 9 |
| High school diploma or equivalent | 16 |
| Post-secondary education, no bachelor's | 8 |

**Knowledge of PrEP.** The majority of participants across focus groups indicated that this was their first time hearing about PrEP. Of those that indicated that they were aware of PrEP, only a couple knew that PrEP was an HIV prevention method. As one participant explained it: *"[PrEP] is sort of a preventive measure to help you not become infected with the HIV virus. And if you've already had the virus I don't think you're supposed to take it"* (Community 2*).* Several other participants had heard of PrEP but only in passing, mostly from television commercials. Other sources where participants had heard of PrEP included previous HIV prevention studies in the community, the health department, a doctor, the radio, a previous focus group, relatives, and at work. The television commercials that women had seen about PrEP seemed to indicate to them that PrEP was for men and not for women: *"I thought it was only for men. Mostly commercials are saying it's just for men"* (Community 3). This had led some to believe that women could not use PrEP: *"Because at first when they started talkin' about PrEP, it was only for men. I personally didn't know that it was for women as well"* (Community 1). Participants also shared that they thought women in their community did not have knowledge about PrEP, primarily due to lack of available education: *"I don't think they're educated [about PrEP]"* (Community 3).

**Attitudes toward PrEP.** After discussing preexisting PrEP knowledge, participants were shown a brief CDC video explaining PrEP. Following this video, participants expressed mostly positive perspectives on PrEP but held a wide range of attitudes. Participants also expressed curiosity about PrEP and the desire to learn more about it. Participants were mostly positive about the potential benefits of PrEP: *"Anybody having sex should use it, period!"* (Community 2). Some emphasized the benefits of PrEP, particularly for young people, who they perceived as more likely to be sexually active and less cautious:

> *"I think it would make a big difference with the younger generation that are not as cautious about bein' sexually active to prevent them from havin' somethin' long term that may kill them..."* – Community 1

Participants also largely agreed that women in their community would be interested in using PrEP, citing the desire to mitigate concerns about disease transmission and mortality. They emphasized the necessity of PrEP in overcoming reliance solely on a partner's word regarding their health status. One participant expressed:

*"Cause they don't have to worry about catching no disease or dying… you know you ask a man 'Do you have such and such,' 'No I don't have it,' Yeah, right. This way you don't have to worry about that."* – Community 2

Some participants expressed that anyone who is sexually active should use PrEP, seeing it as a protective measure that many women could benefit from regardless of their reasons for HIV prevention. For example, this participant's perspective was informed by the experience of her mother who had passed away from AIDS-related complications:

*"Yeah, I think that would be good, too, for everyday women 'cause you never know who you dealin' with. Yeah, cause you never know what your mate, you don't. Someone you messin' with that would have it… [my mother] would still be here today, she goin' to work, mindin' her business, payin' her bills, and tryin' to take care her family. She didn't even know."* – Community 1

Other participants focused on specific populations they believed could benefit from PrEP, including sex workers and individuals using drugs, saying: "*Yeah, I think it's a good thing, too, for prostitutes because they out here, and you don't know what they doin'. Then, too, you said the drug addicts. You never know what they doin'…*" (Community 1). Some participants viewed that PrEP would be beneficial for individuals with multiple sexual partners, especially for those who do not know the HIV status of their sexual partners:

*"… Everybody in [this city] need to be on it because everybody—well me and you in the same boat, but everybody sleeping with everybody. You don't know if somebody you done slept with got it or not. Got it from somebody—you coulda got it from her and didn't say nothin', 'cause she nothin' and there's people out here that got it, and they ain't saying nothin'"* – Community 4

Though women were generally positive about PrEP and identified types of women they believed could benefit from PrEP, a few specifically expressed that they were not interested in PrEP because they did not perceive themselves to have a high likelihood of acquiring HIV. For some this was because they were not sexually active: "*Well, I ain't having sex, so I ain't worrying about catching nothin*" (Community 4).

Some participants did not believe PrEP would be helpful to women in their community. For example, one participant mentioned that there was nothing special about PrEP because people using it would still need to use other forms of protection.

*"With all the other traditional precautions to be taken, I don't see the point of taking the pill, I think you just did. You know, if you happen to use the protection and other stuffs [condoms], I feel like a lot of people would feel like what is the point, you know"* – Community 2

Other participants highlighted that, when considering PrEP versus condoms, some women may opt for PrEP due to allergies or a general aversion to condom use. The perceived sense of guarantee and relief offered by PrEP, especially for individuals facing societal pressure or criticism from friends and family regarding condom use, was emphasized:

*"Yeah, and some of them just don't use [condoms], period. They love that 'cause that's even gonna help them even better to guarantee they know—I ain't gotta take no rubbers—'cause*

*her friends and everybody cussin' at her, her mama, and sayin', 'You better wear them rubbers'… [with PrEP] they ain't gotta worry about it no more."* – Community 1

**PrEP questions and concerns.** After being shown the informational video about PrEP, common questions and concerns that women had about PrEP focused on the side effects of PrEP. One participant specifically described her worries about side effects to the kidneys, saying: *"That whole conversation was going great until you said kidney. That would throw me off, you know… cause once your kidney starts acting up you're never gonna act up,"* (Community 3). Other participants wondered about mixing PrEP with other medications: *"What about the side effects when you're already on medication?"* (Community 2).

Some participants were concerned about the burden of PrEP, describing the daily dose frequency requirements of PrEP as something some individuals might struggle to adhere to, especially those who lead busy lives and do not like taking pills. Some viewed that PrEP would be better as a weekly pill or injection.

*"If there was one pill a week you could take or… a shot, I think it would be more effective 'cause it's like, 'Let's do it, get it over with,' and they don't have to think about it every day. 'Oh, did I take my pill? Oh, I forgot.' It'll just be, 'Let me take this one pill. That last me this week. I ain't got to really think about that again until next week.' Or the shot, however many months it may last or if it was just once a month, I think it'll be a little more effective."* – Community 1

Other questions and concerns raised included the size and difficulty of swallowing PrEP pills, if women could take PrEP while pregnant, and the dosage of PrEP. Others had questions about payment: *"Is it gonna be available for people with certain insurance? Will it be available to people without insurance? Is it affordable?"* (Community 3).

**Anticipated community PrEP perceptions.** In addition to their personal perspectives, we asked participants to reflect on how they anticipate other women in their community might view or be interested in PrEP. Overall, opinions were mixed. One participant explained:

*"I ain't even gonna lie. Some of them be like, 'That's too much. I ain't got time for that.' Then some of them might actually be interested in it. It's like, yeah, if it's a chance you might get some, you might not get some"* – Community 4

Several women believed that their peers who are sexually active may be interested in PrEP: *"I can't speak for the next person, but if they have an open mind, and they are having sex that this would be something that they would want to take"* (Community 3). Others anticipated that most women would be reluctant to use PrEP, but this could be overcome with education: *"I think they'll be reluctant at first, but it's all about education, too, and it's all about who talking to you about it"* (Community 1).

Some participants shared that women in their communities would view PrEP as an indication of taking care of oneself: *"Overall, women would feel good about it because they know they're taking care of themselves and you would feel good about it if you were to"* (Community 4). On the other hand, some participants suggested that women in their communities may not be inclined to use PrEP because of the burden of the daily dose, low perceived effectiveness of PrEP, concern about potential side effects, medical mistrust in the community, a general disinterest in PrEP, and the perceived distrust it could introduce in a romantic relationship: *"I guess a lot of women wouldn't want to. Cause the man is like 'you ain't supposed to be doing that* [having other sexual relationships] *with nobody no way"* (Community 2).

Some discussed how women's interest in PrEP might be affected by HIV-related stigma if a woman taking PrEP was perceived to be living with HIV: *"I think some of them just think it's a stigma. Even though you don't have it, the fact that you're takin' it, now, somebody's gonna associate you with havin' it. As opposed to it bein' a preventive, they're lookin' at it like, 'Somebody gonna think I have it or was messin' with somebody that had it'. You know what I'm sayin'?"* (Community 3). Other participants shared how misinformation or misconceptions about PrEP could perpetuate stigma-related barriers. One participant shared her personal experience with misconceptions that led to her initial dismissal of PrEP as a measure exclusively for individuals already diagnosed with HIV:

*"When I saw [PrEP] sittin' in there—I just got my STDs things done, and that lady did a little stuff, and I seen your PrEP sign in the health department, in the office. It's the AIDS prevention. Then somebody was like, "No, girl. That's just if you got it… Then I didn't ask no more questions about it. They was just like, somebody on the street was, "No, that's just if you got it. It makes it don't look like you got it."* – Community 3

Other women discussed the medical mistrust that could prevent women from wanting to use PrEP. Specifically, women in their community might feel as though PrEP is too new and that they are being used to test the medication: *"Or because it's not out here enough, they might feel like they the person gettin' researched or studied on"* (Community 1).

In one focus group, religious beliefs also emerged as a potential influence on community perceptions regarding PrEP, with certain faiths discouraging protection or contraception, potentially dissuading individuals from using PrEP. One participant said:

*"Some religions don't believe in uh married people using protection or something other than nature. It's a sin, you're not supposed to use nothing 'cause you're supposed to be married and only be with each other so a lot of people probably wouldn't, a lot of religious people probably wouldn't use it…. Because you're supposed to do it the righteous way, you push in I push out that's it no more extra, nothing so."* – Community 2

Other practical barriers were raised regarding women's ability to access PrEP. These included affordability concerns, with participants explicitly stating their inability to afford such medications, and the discussion delved into individual-level barriers such as willingness to use PrEP and daily adherence. Transportation challenges were also highlighted as a potential hindrance, particularly for individuals without vehicles, impacting access to PrEP-related appointments and medication:

*"A lot of people don't have vehicles and stuff, so a way for people to get to their appointments and stuff, or wherever they need to get to take the medication. Find a way for them to get there to take the medication. That means they need transportation there and needs for transportation to get them back."* –Community 4

**Recommendations to promote community PrEP awareness and interest.** We also asked women to reflect on ways to promote accurate PrEP awareness in the community. Overall, participants emphasized early education, starting discussions about sexual health and PrEP in middle school, as important to better inform individuals about PrEP. Concerns were raised about the gender-specific presentation of PrEP in commercials, advocating for a gender-neutral approach.

Participants repeatedly discussed education as a key facilitator for PrEP uptake, recommending multiple channels of communication including TV ads, flyers, and online forums for

PrEP education. One participant mentioned that: "[Women in the community] *just need to be educated. I mean, I don't see anything else, I think that we need to hear speakers talk about it or set up booths maybe, and distribute flyers*" (Community 2). Some participants emphasized the need for early education, incorporating discussions about sexual health and PrEP in middle school.

> "*If they would talk about it where you talk about sexual health in middle school. If you started, then, talkin' to girls or guys about it, then you've been hearin' about it for years. You're better informed all through middle school, high school… And know the options*" –
> Community 1

## Discussion

Among the women who participated in our survey and focus group discussions, few had previous awareness or knowledge of PrEP, and very few knew anyone who had used it. TV commercials and health care providers were the most common sources of PrEP knowledge, with TV commercials leading women to think it could only be used by men. Our findings are similar to other studies that have found women's knowledge of PrEP in the United States to be generally low, with only up to a third of women having heard of it [49].

Most women in our study expressed positive attitudes toward PrEP and an interest in using it if affordable. High levels of PrEP interest have been found in other recent studies with cisgender women in the United States but actual uptake may be significantly lower; only 11% of women participating in a PrEP navigation program ultimately initiated PrEP [50]. Despite high interest, side effects, cost, and daily dosing were common concerns we heard from women, similar to findings in previous studies [31–33,51,52]. Transparent messaging regarding side effects and the proportion of users experiencing each side effect is necessary both on ethical grounds and to address concerns regarding more severe yet less common side effects. With regard to daily dosing, messaging should provide information on both daily and upcoming long-acting PrEP methods, including injectable medication, and emphasize the importance of personal choice to select a long-acting method if preferred. Helping women interested in PrEP understand the potential cost is also important and complex given differing insurance statuses and coverage, diverse payment support in varied care contexts, and ever-evolving federal policy regarding coverage of new PrEP methods as a preventive care service. To address cost considerations, information regarding all available PrEP access points both public and private, local and virtual should be provided, including telehealth platforms through which women may access PrEP at no cost. While the majority of our participants were Medicaid or Medicare recipients, 20% were uninsured which may have affected cost-related perceptions. North Carolina has recently adopted Medicaid expansion since the time of this study, which may reduce cost-related concerns and increase PrEP access among women [53].

Other studies with Black or African American women found a general interest in PrEP but specific concerns about stigma [32,54–57]. While most women we surveyed believed that family, partners, and friends would approve if they decided to use PrEP, only half thought their other community members would approve. Women in focus groups also identified stigma as a potential barrier to community interest in PrEP. They noted that PrEP's association with HIV might lead to confusion, causing people to believe it is intended only for those living with HIV, and that this misconception could hinder PrEP uptake. Evidence from other US studies indicates that stigma significantly impacts women's interest in PrEP, acting as a barrier to uptake [32,54–57]. This is evident in the fear of being perceived as promiscuous, HIV-positive, or homosexual, and the expectation of disapproval from family, partners, and friends [4,58,59].

Based on these findings, addressing PrEP stigma and misunderstandings of PrEP as being for people living with HIV is crucial to promote PrEP acceptance and uptake among women.

Perceived HIV acquisition chances were also an important consideration in women's interest in PrEP in our study. While some women in our focus groups believed that PrEP would be valuable to offer protection and peace of mind regardless of specific reasons for HIV prevention, others viewed that PrEP would only be appropriate for women in their community with clear reasons for HIV prevention, such those who were sex workers, used substances, or had multiple sex partners. In our survey, more than half of women reported at least one HIV-associated factor, but only 10% reported a moderate or high perceived chance of HIV acquisition. Yet, a high proportion of women both with and without any reported factors were interested in using PrEP (though more women with reported HIV-associated factors were interested in PrEP than women who reported no factors). This combination of low perceived chance of HIV acquisition yet high interest in PrEP has been observed among women in other settings [60], as women (including those in our focus groups) may be motivated by the peace of mind provided by PrEP even though they may not perceive themselves to have a high chance of HIV acquisition. Yet, past studies suggest that underestimation or underreporting of HIV-associated factors is a barrier to PrEP uptake for women [3,59,61,62]. In general, women perceive lower personal chances of HIV acquisition than men [63,64], and may underestimate their chances because they are not aware of their partner(s)' sexual history [65,66]. Some women may underrate their chances of acquisition even in the context of known partner non-monogamy or serodifferent relationships [67]. However, lower perceived chances of acquisition may not translate to disinterest in PrEP [60]. Some women may perceive low chances of HIV acquisition but be interested in PrEP as a source of sexual health empowerment [68]. More studies on the impact of these perceptions on PrEP uptake are needed.

Other important considerations which arose among women in our study were the dose burden of daily oral PrEP and dual protection with condoms. Some participants suggested that a longer-acting pill or injectable medication would be more appealing to women in the community than a daily pill. Research on women's preferences for oral versus injectable PrEP in the United States reveals a range of preferences [69–71], with women's preference potentially mirroring their personal preference for birth control modality [72]. While the other results of our study primarily pertain to daily oral PrEP, concerns related to side effects, cost, and perceived need for HIV prevention are applicable to long-acting PrEP methods. Women in our study also held mixed opinions about the need for dual protection with condoms to prevent other STIs if using PrEP, with some viewing dual protection to be necessary and others viewing that the perceived need for dual protection could dissuade women from using PrEP. Women's preferences for dual protection were also mixed in other studies [31,70]. Reasons that women may want to use PrEP alone without condoms may include the inability to negotiate condom use with a sexual partner, such as in the context of intimate partner violence [73,74], or disliking the experience of using condoms [70]. In both cases of method choice and use of dual protection, accurate counseling on the risks and benefits of each option to facilitate person-centered choice is needed.

To address PrEP awareness and promote uptake, women in our study recommended education through multiple channels that is gender-neutral so it is clear that cisgender women can use PrEP. They also recommended early education starting in middle school on PrEP and emphasized that both the relatability of the person or source delivering messaging would be critical to promote trust in the information provided. Currently, few approaches to raise PrEP awareness among women in the United States have been tested. Those to date include a salon-based intervention [75–77], and social media campaigns [78,79]. Diversified interventions to reach important sub-populations of cisgender women – including those living in low-income

communities – are needed. While the results of this study primarily have implications for interventions aimed at improving community knowledge and attitudes, other strategies to address social determinants of PrEP uptake (including health care access, low insurance coverage, and community HIV stigma [19–27]) are needed. The results of this study have been used to inform a community-awareness raising campaign delivered in complement to a mobile PrEP clinic program aiming to address the key determinant of health care access.

## Limitations

The results of this study should be interpreted with key limitations in mind. First, survey participants were recruited on a convenience basis, thus results may not represent the experiences and opinions of all women living in the included public housing communities. To encourage representation (albeit not through probabilistic methods), we conducted outreach to most public housing communities in the sampling frame on repeated occasions and varied our recruitment locations and times within each community to recruit as broadly as possible. Second, participant responses to questions regarding HIV-associated factors and PrEP perceptions are subject to social desirability bias. To mitigate this bias, the survey was completed through participant self-administration, and assurances were provided regarding data anonymity and privacy. Focus group participants were reminded throughout the discussion that there were no right or wrong answers to the questions posed and were provided information on how the study team would preserve data privacy and anonymity. Third, participants were allowed to skip survey questions they did not want to answer on ethical grounds, and up to 17% of responses were missing on some items. For this reason, potential biases in the results regarding the sensitivity of more commonly skipped questions should be considered; for example, participants who did not report their household income may have chosen not to do so if they perceived their income to be an outlier in the community. Fourth, associations should not be interpreted as causal given their bivariate nature and the cross-sectional data utilized; rather, bivariate associations presented are meant to offer descriptive understanding of PrEP perceptions among women with differing levels of reasons for HIV prevention. Fifth, this study does not directly address women's interest in long-acting injectable cabotegravir, as this product was not FDA approved at the time of the study. Finally, the results of this study may not be generalizable to women in other contexts in the US and globally. Given similar themes reported in other studies conducted with women of Black women and in other lower-income communities in the US, our results may have relevance to similar communities in the Southeastern US.

## Conclusions

Women living in public housing communities in a Southeastern city had limited knowledge of PrEP but expressed high interest in using it, despite sometimes perceiving their chances of HIV acquisition as low. Concerns regarding side effects, cost, and daily dosing were identified as key barriers to PrEP uptake among women in the community. Media- and community-based communication efforts that include cisgender women should focus on transparent discussions about side effects, emphasize the importance of personal choice, and provide comprehensive information on all available PrEP methods to increase PrEP interest among women. Messaging delivered by trusted sources (e.g., trusted peers, professionals familiar in the community) may be critical to the success of educational campaigns. PrEP education should prioritize accurate, destigmatized discussions about personal reasons for HIV prevention, and promote person-centered choices among emerging PrEP methods and dual protection. Evidence-based community- and media-based PrEP education programs are needed;

future research should seek to develop and evaluate education programs in varied contexts to promote generalizability and transferability.

## Author contributions

**Conceptualization:** Lauren M. Hill, Marcella Jones, Suur Ayangeakaa, Alexandra F Lightfoot, Mehri S McKellar, Carol E Golin.

**Data curation:** Olivia Allison, Oluwamuyiwa Adeniran, Tonya Stancil, K. Jean Phillips-Weiner.

**Formal analysis:** Lauren M. Hill, Olivia Allison, Oluwamuyiwa Adeniran.

**Funding acquisition:** Lauren M. Hill, Carol E Golin.

**Investigation:** Lauren M. Hill, Marcella Jones, Suur Ayangeakaa, Alexandra F Lightfoot, Mehri S McKellar, Carol E Golin.

**Methodology:** Lauren M. Hill, Marcella Jones, K. Jean Phillips-Weiner, Alexandra F Lightfoot, Mehri S McKellar, Carol E Golin.

**Project administration:** Lauren M. Hill, Tonya Stancil, K. Jean Phillips-Weiner, Mehri S McKellar, Carol E Golin.

**Resources:** Mehri S McKellar.

**Supervision:** Lauren M. Hill, Tonya Stancil, Carol E Golin.

**Writing – original draft:** Lauren M. Hill.

**Writing – review & editing:** Lauren M. Hill, Olivia Allison, Oluwamuyiwa Adeniran, Marcella Jones, Suur Ayangeakaa, Tonya Stancil, K. Jean Phillips-Weiner, Alexandra F Lightfoot, Mehri S McKellar, Carol E Golin.

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
