## [Decision Letter · Decision Letter 0]

3 Oct 2024

PONE-D-24-26120PrEP knowledge and perceptions among women living in North Carolina public housing communitiesPLOS ONE

Dear Dr. Hill,

Thank you for submitting your manuscript to PLOS ONE. After careful consideration, we feel that it has merit but does not fully meet PLOS ONE’s publication criteria as it currently stands. Therefore, we invite you to submit a revised version of the manuscript that addresses the points raised during the review process.

**Please address the comments from the reviewers in full prior to resubmission.**** ** ==============================

We look forward to receiving your revised manuscript.

Kind regards,

Douglas S. Krakower, MD

Academic Editor

PLOS ONE

2. In the online submission form you indicate that your data is not available for proprietary reasons and have provided a contact point for accessing this data. Please note that your current contact point is a co-author on this manuscript. According to our Data Policy, the contact point must not be an author on the manuscript and must be an institutional contact, ideally not an individual. Please revise your data statement to a non-author institutional point of contact, such as a data access or ethics committee, and send this to us via return email. Please also include contact information for the third party organization, and please include the full citation of where the data can be found.

Reviewers' comments:

Reviewer's Responses to Questions

**Comments to the Author**

1. Is the manuscript technically sound, and do the data support the conclusions?

Reviewer #1: Yes

Reviewer #2: Yes

2. Has the statistical analysis been performed appropriately and rigorously? 

Reviewer #1: Yes

Reviewer #2: Yes

3. Have the authors made all data underlying the findings in their manuscript fully available?

Reviewer #1: Yes

Reviewer #2: Yes

4. Is the manuscript presented in an intelligible fashion and written in standard English?

Reviewer #1: Yes

Reviewer #2: Yes

5. Review Comments to the Author

Reviewer #1: Research focused on Black women and PrEP is of priority for reaching the End the HIV epidemic 2030 goals. This paper provides insights from Black women in the US south and can serve as a basis for how best to engage women in HIV prevention and improve PrEP uptake. A few items for authors to consider:

1. Line 34 discusses social and economic structures contributing to barriers to PrEP, but there is no mention of structural racism called out. Given the long standing literature that supports the impact of racism on health and health outcomes, calling it out is warranted.

2. There are a few items in the discussion section that can be added such as implications for the new forms of PrEP delivery for this population, the influence of policy being that majority of participants had Medicaid, and the report of one partner by majority of the women- what influence do men have in PrEP education and uptake for this population.

3. Consider the use of language around "risk" See Risk to Reasons publication by Viv: https://viivhealthcare.com/content/dam/cf-viiv/viivhealthcare/en_US/pdf/from-risk-to-reasons-reframing-hiv-prevention-and-care-for-black-women-spreads.pdf

4. Finally, there is a link to social determinants of health (SDOH) and health outcomes, PrEP use and acquisition of HIV- many of which were discussed in the background, however, limited data collected about SDOH to examine how these intersect with the findings reported. There is an opportunity to mention in the discussion as a future need and/or a limitation so that the science can shift to addressing and examining SDOH in a way that interventions and programs will reflect strategies that respond to these SDOH.

Thank you for the opportunity to review this manuscript and thank you for your work in prioritizing the needs of Black women in HIV prevention.

Reviewer #2: REVIEWER RESPONSE

PONE-D-24-26120

An Integrated Intervention to reduce HIV risk behaviors among heterosexual HIV- negative, PrEP knowledge and perceptions among women living in North Carolina public housing communities

Reviewer Comments:

The study offers insights into the lack of awareness and favorable opinions regarding PrEP, among women from disadvantaged areas. Specifically African American women, in the Southeastern United States region. Although the research impressively combines methodologies and sheds light on discoveries related to obstacles facing PrEP adoption there are certain aspects that could benefit from enhancements in terms of clarification, thoroughness and overall impact. There are opportunities, for improvement listed below that will help enhance its clarity, coherence and academic value.

Abstract:

Strengths:

• The abstract clearly outlines the problem (lack of awareness of PrEP among women in low-income communities).

• The objective of the study is well-defined, focusing on women’s knowledge and perceptions of PrEP.

• The results provide valuable insights into both quantitative and qualitative data.

Suggestions for Improvement:

• Clarity on Specifics: The abstract could benefit from a more specific presentation of key statistics. For instance, the mention that “35% had heard of PrEP” and “61% would take it” can be highlighted further with brief explanations of their significance.

• Problem Context: Add a sentence that frames the problem more globally or within the wider literature on PrEP uptake in underserved populations to emphasize the importance of the findings.

• Conclusion: The final sentence could offer a stronger closing message about the implications of these results for future PrEP outreach and intervention strategies.

Introduction:

• It would be helpful to more clearly highlight gaps in the existing literature, especially in PrEP campaigns targeted at women in low-income communities.

• The introduction could flow better with more transition sentences linking various sections (e.g., from HIV prevalence to why PrEP remains underutilized).

• A section briefly reviewing perceived barriers such as stigma, cost, and misinformation in the introduction would provide context for the later findings.

Methodology:

• The sampling is described as convenience-based. A more detailed explanation of why this method was chosen, along with its potential limitations (e.g., selection bias), should be provided.

• Address any potential limitations with respect to non-responses or how different sociodemographic factors were controlled for in the analysis.

Results:

• Tables are somewhat dense and could be simplified or split into more digestible sections. Key findings from the tables could also be highlighted or summarized in the text for easier reading.

• The qualitative insights from focus groups could benefit from more examples or direct quotes to bring participant perspectives to life.

• The manuscript mentions differences between women with HIV risk factors and those without but could delve deeper into explaining these differences (e.g., using subgroup analyses).

Discussion:

Suggestions for Improvement:

• The discussion would benefit from more direct references to other studies on PrEP uptake in similar populations. This could reinforce the validity of the study’s findings.

• Since side effects and costs are recurring concerns, more emphasis should be placed on how future campaigns can specifically address these barriers.

• The limitations are not deeply discussed. While convenience sampling is mentioned, issues such as the generalizability of the findings to women outside public housing should be acknowledged.

Citations:

• Ensure the inclusion of the most recent studies in the field of PrEP uptake and women’s health to keep the research current.

• Consistency: Double-check for citation consistency, especially in terms of formatting. All references should be thoroughly proofed to follow journal guidelines.

Conclusion:

• Add more actionable takeaways or specific next steps that could guide future researchers, practitioners, or policymakers in PrEP intervention design.

6. PLOS authors have the option to publish the peer review history of their article (what does this mean? ). If published, this will include your full peer review and any attached files.

**Do you want your identity to be public for this peer review?** For information about this choice, including consent withdrawal, please see our Privacy Policy .

Reviewer #1: No

Reviewer #2: **Yes: ** Dawn Goddard-Eckrich

---

## [Author Response · Author response to Decision Letter 1]

5 Feb 2025

Response to Reviewers

We thank the reviewers for their thoughtful comments on our manuscript. We have revised the manuscript in response to these comments as summarized below.

Reviewer #1

1. (Introduction) Social and economic structures as barriers to PrEP is discussed but there is no mention of structural racism called out.

Response: We have added a statement addressing the role of structural racism in shaping social determinants of health and the health outcomes (beginning line 40 of the Introduction).

2. (Discussion) More details could be added to the discussion with regard to: a) the implications of the results for new forms of PrEP; b) the influence of policy for the study population primarily made up of Medicaid recipients; c) the influence of men on PrEP education and uptake on women in this population.

Response: We have addressed each suggestion as follows: a) We have added a statement regarding the implications of our results for long-acting PrEP methods (line 469). The preceding part of the same paragraph also addresses our direct results relating to long-acting PrEP; b) We have added discussion of the potential impact Medicaid policy in the study setting (beginning on line 425); c) Our results do not directly address the role of men in women’s PrEP education, thus we have chosen not to include this in the Discussion.

3. Re-consider the use of language around "risk."

Response: We have replaced this term throughout the manuscript and provide examples of the revisions below, following the guidance from ViiV cited by the reviewer (https://viivhealthcare.com/content/dam/cf-viiv/viivhealthcare/en_US/pdf/from-risk-to-reasons-reframing-hiv-prevention-and-care-for-black-women-spreads.pdf) and NIAID (https://www.niaid.nih.gov/sites/default/files/niaid-hiv-language-guide.pdf) wherever possible.

Old term New term

HIV risk factors HIV-associated factors

Risk perception perceived HIV acquisition/exposure chances

Women with HIV risk indications Women with reasons for HIV prevention

4. There is an opportunity to mention in the discussion a future need for research to shift to addressing and examining social determinants of health in a way that interventions and programs will reflect strategies that respond to these SDOH.

Response: We have added statements to the Discussion (beginning line 486) highlighting the importance of addressing not only knowledge and attitudinal factors but also social determinants of PrEP access. We further provide context by sharing that the results of this study were used to develop an awareness-raising campaign that complemented a mobile PrEP clinic program, aimed at addressing the social determinant of health care access.

Reviewer #2

1. (Abstract) The abstract could benefit from further interpretation of key statistics presented.

Response: We have clarified the significance of key results presented in the abstract, including an interpretation of the results regarding the fact that few women had previous knowledge of PrEP but despite this many were interested in it and expected positive community attitudes toward PrEP.

2. (Abstract) The final sentence could offer a stronger closing message about the implications of these results for future PrEP outreach and intervention strategies.

Response: We have strengthened the concluding remarks in the abstract to offer more specific recommendations for future PrEP outreach, including the need to specifically address cisgender women, provide transparent information on side effects, and offer destigmatized messaging about why women may consider PrEP use.

3. (Abstract) Add framing of the problem addressed by this paper in a broader context of PrEP uptake in underserved populations to emphasize the importance of the findings.

Response: Due to word limit constraints, we were unable to add this broader context in the abstract but address this in the Introduction, where we discuss PrEP uptake among cisgender women in the national context.

4. (Introduction) It would be helpful to more clearly highlight gaps in the existing literature, especially in PrEP campaigns targeted at women in low-income communities.

Response: While there have been online campaigns targeting cisgender women in the US, to our knowledge no evidence-based approaches for community-level communication targeting cisgender women are currently available, but one is in development (line 60).

5. (Introduction) The introduction needs more transition sentences linking various sections.

Response: We have reviewed the Introduction and added transitional phrases to improve flow and clarity.

6. (Introduction) Address specific PrEP uptake barriers such as those in the findings in the introduction to provide context for the results.

Response: We have added a discussion of PrEP uptake barriers related to side effects, cost, and stigma to the Introduction (line 45).

7. (Methods)The sampling is described as convenience-based. A more detailed explanation of why this method was chosen, along with its potential limitations should be provided.

Response: We have added our rationale for convenience-based sampling in the Methods section (line 85). In the Limitations section, we discuss the potential biases associated with the sampling approach. Convenience sampling was selected due to unavailability of a resident list or contact information for probabilistic sampling. Survey participants were recruited on a convenience basis, thus results may not represent the experiences and opinions of all women living in the included public housing communities. To encourage representation (albeit not through probabilistic methods), we conducted outreach to most public housing communities in the sampling frame on repeated occasions and varied our recruitment locations, times, and channels within each community to recruit as broadly as possible.

8. (Methods) Address any potential limitations with respect to non-response or how different sociodemographic factors were controlled for in the analysis.

Response: We have added a statement in the Limitation sections regarding survey non-response (beginning line 502). Because our analyses did not include causal or multivariate models, no controls were included. Rather, we present descriptive statistics and bivariate comparisons to offer descriptive understanding of PrEP perceptions among women with differing reasons for HIV prevention. We have added this caveat to the Limitations (beginning line 507).

9. (Results) The tables are dense and could be simplified or split into more digestible sections. Key findings from the tables could also be highlighted or summarized in the text.

Response: To make the tables less dense, we made the following changes. In Table 1 and Table 7, we collapsed some of the categories reported in the descriptive statistics for digestibility. We split Table 2 into two separate tables (distinguishing between results regarding HIV-associated factors and personal HIV perceptions). Key findings from the tables are also summarized in the text.

10. (Results) More quotes would help bring the participant perspectives from the focus groups to life.

Response: We have ensured that each major finding from the focus groups is supported by at least one participant quote.

11. (Results) The manuscript mentions differences between women with HIV-associated factors and those without but could delve deeper into explaining these differences (e.g., using subgroup analyses).

Response: These analyses are outside the scope of the current paper in which our goal is to characterize a broad range of constructs related to PrEP interest and perceptions in the study population. We are exploring the feasibility of future analyses to better understand subgroup differences for a future publication.

12. (Discussion) Augment the discussion of PrEP uptake in similar populations to contextualize study findings.

Response: There is limited information on PrEP uptake among similar populations, but we have added available statistics to provide context to the Discussion (line 419).

13. (Discussion) Since side effects and costs are recurring concerns, more emphasis should be placed on how future campaigns can specifically address these barriers.

Response: We have added recommendations to address concerns around side effects as well as the cost of PrEP and associated care to the Discussion (beginning line 422). These recommendations include transparency in side effect information and clarification on the commonality of each side effect, and information on diverse PrEP access points which may have different implications for cost (e.g. public vs. private providers, local vs. telehealth options).

14 (Discussion) Details should be added to the Limitations regarding the generalizability of the findings to women outside public housing.

Response: We have added a statement to the Limitations regarding the potential inability to generalize the findings to other populations and settings in the US and globally and offered suggestions for the potential relevance of our results to similar populations and communities in the Southeastern US (line 511).

15. (Conclusions) Add more actionable takeaways or specific next steps that could guide future researchers, practitioners, or policymakers in PrEP intervention design.

Response: We have added specific recommendations to the Conclusions highlighting features critical to the success of PrEP education campaigns based on our results, such as delivery by trusted sources and an emphasis on choice. We also outlined actionable recommendations for future interventions to promote PrEP awareness and access among cisgender women.

16. (Citations) Ensure the that the most recent relevant studies on PrEP uptake among women are included.

Response: We have updated the literature review and added citations for relevant studies published in the past three years.

17. (Citations) Double-check for citation consistency, especially in terms of formatting. All references should be thoroughly proofed to follow journal guidelines.

Response: We have edited the in-text citations and references cited to ensure formatting consistency and compliance with journal guidelines.

Additional changes made in response to Journal Requirements:

We have addressed journal requirements as follows:

• We have reviewed journal style guidelines and have ensured that our title page, manuscript, and references meet PLOS ONE's style requirements.

• We have updated our data statement to indicate an institutional, non-author point of contact for data access.

• We have reviewed our reference list to ensure that it is complete and correct. Specifically, we have added references to bring our literature review up to date and to address new points in the Introduction and Discussion (per responses to reviewers above). We have also corrected in-text citation and reference formats to meet journal formatting standards.

---

## [Editor Report · Decision Letter 1]

13 Feb 2025

PrEP knowledge and perceptions among women living in North Carolina public housing communities

PONE-D-24-26120R1

Dear Dr. Hill,

We’re pleased to inform you that your manuscript has been judged scientifically suitable for publication and will be formally accepted for publication once it meets all outstanding technical requirements.

Kind regards,

Douglas S. Krakower, MD

Academic Editor

PLOS ONE
---

## [Editor Report · Acceptance letter]

PONE-D-24-26120R1

PLOS ONE

Dear Dr. Hill,

I'm pleased to inform you that your manuscript has been deemed suitable for publication in PLOS ONE. Congratulations! Your manuscript is now being handed over to our production team.

Kind regards,

on behalf of

Dr. Douglas S. Krakower

Academic Editor

PLOS ONE